# Growth and Characterisation Studies of Eu_3_O_4_ Thin Films Grown on Si/SiO_2_ and Graphene

**DOI:** 10.3390/nano11061598

**Published:** 2021-06-17

**Authors:** Razan O. M. Aboljadayel, Adrian Ionescu, Oliver J. Burton, Gleb Cheglakov, Stephan Hofmann, Crispin H. W. Barnes

**Affiliations:** 1Cavendish Laboratory, Physics Department, University of Cambridge, Cambridge CB3 0HE, UK; ai222@cam.ac.uk (A.I.); cheglakovg@yahoo.co.uk (G.C.); chwb101@cam.ac.uk (C.H.W.B.); 2Diamond Light Source, Didcot OX11 0DE, UK; 3Department of Engineering, University of Cambridge, Cambridge CB3 0FA, UK; ob303@cam.ac.uk (O.J.B.); sh315@cam.ac.uk (S.H.)

**Keywords:** Eu_3_O_4_, graphene, thin film, heterostructure, metamagnetism, XPS

## Abstract

We report the growth, structural and magnetic properties of the less studied Eu-oxide phase, Eu${}_{3}$O${}_{4}$, thin films grown on a Si/SiO${}_{2}$ substrate and Si/SiO${}_{2}$/graphene using molecular beam epitaxy. The X-ray diffraction scans show that highly textured crystalline Eu${}_{3}$O${}_{4}$(001) films are grown on both substrates, whereas the film deposited on graphene has a better crystallinity than that grown on the Si/SiO${}_{2}$ substrate. The SQUID measurements show that both films have a Curie temperature of ∼5.5±0.1 K, with a magnetic moment of ∼320 emu/cm${}^{3}$ at 2 K. The mixed valence of the Eu cations has been confirmed by the qualitative analysis of the depth-profile X-ray photoelectron spectroscopy measurements with the ${\mathrm{Eu}}^{2+}$:${\mathrm{Eu}}^{3+}$ ratio of 28:72. However, surprisingly, our films show no metamagnetic behaviour as reported for the bulk and powder form. Furthermore, the microscopic optical images and Raman measurements show that the graphene underlayer remains largely intact after the growth of the Eu${}_{3}$O${}_{4}$ thin films.

## 1. Introduction

Mixed-valence or fluctuating valence behaviour are usually found in lanthanide-based compounds due to the intermixing of the *s*−*d* band with the localised *f* band near the Fermi level. Therefore, they exhibit unique magnetic, thermal and electrical properties [1]. Eu cations in Eu-based compounds mostly occur in the 2^+^ valence. However, in trieuropium tetroxide (Eu${}_{3}$O${}_{4}$), Eu ions exhibit a mixed valence of Eu${}^{2+}$ and Eu${}^{3+}$.

Eu${}_{3}$O${}_{4}$ crystallises into an orthorhombic structure (space group *Pnma*) similar to CaFe_2_O_4_ with the lattice parameters *a*
=10.085 Å, *b*
=3.502 Å and *c*
=12.054 Å [2,3]. Figure 1 shows the Eu${}_{3}$O${}_{4}$ structure, where the Eu${}^{2+}$ and Eu${}^{3+}$ ions occupy the Ca${}^{2+}$ and Fe${}^{3+}$ sites, respectively. The oxygen ions form a six- and eight-fold coordination around the Eu${}^{3+}$ and Eu${}^{2+}$ ions, respectively. The coordination is then completed with two oxygen ions lying at the corner of the crystal [2,4]. Therefore, Eu${}_{3}$O${}_{4}$ has two Eu${}^{3+}$ ions and one Eu${}^{2+}$ ion per unit formula [4].

The Eu${}_{3}$O${}_{4}$ compound has an antiferromagnetic arrangement below 5 K. Its bulk and powder forms show a metamagnetic behaviour below the Néel temperature (T_N_), at a critical field of 2.4 kOe [4,5]. A large magnetocaloric effect has also been measured in Eu${}_{3}$O${}_{4}$. Therefore, it is considered a potential material for magnetic refrigeration applications [3]. Although Eu${}_{3}$O${}_{4}$ is a mixed-valence compound, its magnetic ordering is mainly determined by the Eu${}^{2+}$ ions at low-temperature due to the high magnetic moment of Eu${}^{2+}$ (total angular momentum, *J*
=7/2) in comparison to the Eu${}^{3+}$ ions (*J*
=0) [3,4,5]. It has been proposed that the nearest neighbouring Eu${}^{2+}$ ions are strongly coupled by ferromagnetic interactions at low temperature, whereas the distant ions are coupled by weaker antiferromagnetic coupling, resulting in the overall antiferromagnetic state of Eu${}_{3}$O${}_{4}$ [5].

Graphene is a potential material for many technological applications due to its attractive properties, such as its long spin-diffusion length, negligible spin-orbit coupling and high electron mobility [7,8,9]. So far, no study has been reported on the graphene/Eu${}_{3}$O${}_{4}$ system, unlike the well-studied Eu-oxide phase (EuO) where many theoretical and experimental works have been published on the graphene/EuO heterostructure [10,11,12,13,14,15,16,17]. This may well be due to the difficulty of growing Eu${}_{3}$O${}_{4}$, which is the unstable high-temperature phase of Eu-oxide, whereas EuO is grown at room temperature (RT). Eu-chalcogenides and Eu-oxides attracted great attention as promising materials for magneto-optical device applications [18,19,20]. Among these materials, a special interest was given to EuO due to its desirable properties for potential spintronics applications such as spin-filter [14,20,21,22,23,24,25,26]. On the other hand, only a few papers are found on the Eu${}_{3}$O${}_{4}$ compound [2,3,4,5,27,28,29,30,31], making it an interesting metamagnetic compound to be studied as there are still many aspects to be explored on this material.

In this study, 20 nm Eu${}_{3}$O${}_{4}$ thick films were grown on a Si/SiO${}_{2}$ substrate and graphene sheet supported on Si/SiO${}_{2}$ by molecular beam epitaxy (MBE) and subsequently capped with 5 nm of Au. Eu was deposited at high temperatures (300–600 ${}^{\circ }$C) in an oxygen flux. The growth parameters such as the oxygen partial pressure, temperature and deposition rate were optimised to achieve a crystalline Eu${}_{3}$O${}_{4}$(001) phase. The structural characterisation of the films was studied by X-ray diffraction (XRD) and reflection (XRR), where a superconducting quantum interference device magnetometer (SQUID) was used to study their magnetic properties. The results show a successful growth of crystalline, highly textured Eu${}_{3}$O${}_{4}$ (001) films with a Curie temperature (*T*_C_) of ∼5.5 K, which is in agreement with the values reported in References [3,4,5]

Depth-profile X-ray photon electron spectroscopy (XPS) scans were performed to prove the mixed valence of Eu cations in Eu${}_{3}$O${}_{4}$. Furthermore, Raman spectroscopy measurements on the Si/SiO${}_{2}$/graphene/Eu${}_{3}$O${}_{4}$ sample showed that although the growth of Eu${}_{3}$O${}_{4}$ film induces defects in the graphene sheet, the graphene retains its hexagonal lattice structure. Therefore, this study presents one of the first fundamental steps towards understanding the exchange coupling between Eu${}_{3}$O${}_{4}$ and graphene.

## 2. Materials and Methods

Firstly, 20 nm Eu${}_{3}$O${}_{4}$(001) films were deposited on cleaned Si/SiO${}_{2}$ and commercially purchased Si/SiO${}_{2}$/graphene (ACS Material, Pasadena, CA, USA) substrates by a bespoke MBE with a base pressure of 4 × 10${}^{-10}$ mbar. The substrates were heated to 400 ${}^{\circ }$C, while the Eu was evaporated at a rate of 1.2 nm/min. Oxygen was then introduced into the growth chamber, resulting in a partial pressure of 1.1 × 10${}^{-8}$ mbar to deposit Eu${}_{3}$O${}_{4}$(001) at a rate of 1.11 nm/min. A 5 nm film of Au was grown subsequently on the Eu${}_{3}$O${}_{4}$(001) films to prevent them from oxidising to the most stable oxide phase of Eu (Eu${}_{2}$O${}_{3}$). The Au films were deposited at 45 ${}^{\circ }$C, with a rate of 0.057 nm/min at a pressure of 1.9 × 10${}^{-10}$ mbar.

An in-house built quartz crystal microbalance system was used during the deposition to monitor the growth rate and thus the thicknesses of the layers. RT XRD scans were used to study the crystallinity of the grown films. The XRR measurements were performed to confirm the results of the microbalance readings and deduce the density and roughness between the layers. These acquisitions were carried out using a Bruker D8 Discover HRXRD (Bruker, Billerica, MA, USA) with a Cu Kα monochromatic beam with a voltage of 40 kV and a current of 40 mA. The magnetic properties of the Eu${}_{3}$O${}_{4}$ films were studied using a Quantum Design SQUID (Quantum Design, San Diego, CA, USA).

Depth-profile XPS scans were performed on the Si/SiO${}_{2}$/Eu${}_{3}$O${}_{4}$/Au sample using a Thermo Fisher Scientific Escalab 250XiXPS (Thermo Fisher Scientific, Waltham, MA, USA) with Al Kα X-ray source (1486.68 eV, beam width of 500 μm). This was done to study the homogeneity of the Eu${}_{3}$O${}_{4}$ film, determine the atomic ratio of Eu${}^{2+}$ and Eu${}^{3+}$ and confirm the mixed-valence character of the grown film. Furthermore, the effect of depositing Eu${}_{3}$O${}_{4}$ film on the graphene sheet was investigated by Raman spectroscopy measurements using a Renishaw InVia spectrometer (Renishaw plc, Wotton-under-Edge, UK) (100× objective, 10% laser power, spot size of ∼1 μm, 0.5 s exposure time, wavelength of 532 nm). However, the Au capping layer was selectively etched in KI/I${}_{2}$ solution before the measurements to eliminate the Au interference with the Raman measurements. The Si/SiO${}_{2}$/graphene/Eu${}_{3}$O${}_{4}$/Au sample was cleaved into ∼2×2 mm square and placed in the etchant solution for 5 minutes at RT, rinsed with DI water twice, then with IPA and dried with dry N${}_{2}$. Raman scans were then taken at every 50µm in a grid pattern over an area of 1000µm ×1000µm for the region coated with Eu${}_{3}$O${}_{4}$, and every 10µm over an area of 140µm ×140µm for the bare graphene surface.

## 3. Results and Discussion

### 3.1. X-ray Diffraction and Reflection (XRD and XRR)

The RT XRD scans (from 10${}^{\circ }$–100${}^{\circ }$) of the Eu${}_{3}$O${}_{4}$(001) films grown on the Si/SiO${}_{2}$ substrate and on graphene are shown in Figure 2a. The XRD scans show highly textured Eu${}_{3}$O${}_{4}$(002) films with no sign of other oxide-phase of Eu or unreacted Eu within the detection limit of the set-up. Additional Eu${}_{3}$O${}_{4}$(004) and (008) peaks are observed in the scan of the Si/SiO${}_{2}$/graphene/Eu${}_{3}$O${}_{4}$/Au sample, indicating that the underlying graphene layer improves the crystallinity of the Eu${}_{3}$O${}_{4}$ film. This is also proven by the smaller full-width at half-maximum (FWHM) of the Eu${}_{3}$O${}_{4}$ peaks of the Si/SiO${}_{2}$/graphene/Eu${}_{3}$O${}_{4}$/Au sample (0.633${}^{\circ }$) compared to the Si/SiO${}_{2}$/Eu${}_{3}$O${}_{4}$/Au (1.13${}^{\circ }$) as can be seen in Figure 2c. Figure 2b shows the XRR scan and the corresponding fit for the Si/SiO${}_{2}$/Eu${}_{3}$O${}_{4}$/Au sample, whereas the deduced values for the thickness, density and roughness of the layers are listed in the inset table.

### 3.2. SQUID

Figure 3 shows the field-cooled (FC) and zero field-cooled (ZFC) measurements of the Si/SiO${}_{2}$/Eu${}_{3}$O${}_{4}$(001)/Au and Si/SiO${}_{2}$/graphene/Eu${}_{3}$O${}_{4}$(001)/Au samples, respectively. Both show a *T*_C_ of ∼5.5 ±0.1 K, as can be deduced from the d*M*/d*T* vs. *T* (insets), which agrees with the values reported in the literature [3,4,5]. Therefore, care must be given to check for Eu${}_{3}$O${}_{4}$ impurities in EuO${}_{1-x}$ thin films, which sometimes show a pronounced double-dome feature at *T*
<20 K [32,33].

The ZFC isothermal magnetisation measurements as a function of the applied magnetic field for the Si/SiO${}_{2}$/graphene/Eu${}_{3}$O${}_{4}$/Au sample at 2 K, 5 K and 10 K are shown in Figure 4. The hysteresis curves show that the grown Eu${}_{3}$O${}_{4}$ films exhibit ferromagnetic behaviour with a coercive field of 22 Oe. The inset highlights the virgin magnetisation curves at these temperatures. Although the XRD scans (Figure 2a) and the *M* vs. *T* measurements (Figure 3) prove the growth of Eu${}_{3}$O${}_{4}$(001) thin films, surprisingly, the virgin *M*−*H* curves show no metamagnetic transition even with an applied in-plane magnetic field of 3 kOe as reported for crystal and powder Eu${}_{3}$O${}_{4}$ [3,4]. This could be attributed to the strain from the substrate which may be resolved by growing a thicker film.

### 3.3. XPS

The existence of mixed-valence Eu cations was investigated by performing depth-profile XPS scans while measuring the Eu 3*d* and 4*d* spectra simultaneously after Ar${}^{+}$ plasma etching. Figure 5a shows the XPS etch profile of the sample, whereas the XPS survey collected at *t* = 60 s, 360 s and 510 s, highlighting the different detected elements are shown in Figure 5b. The 4*d* XPS spectra have a complicated structure (not shown) due to the strong unfilled 4*f*−4*d* hole interaction, whereas the 3*d* states have a weaker multiplet splitting and broader photoexcitation cross-section. Therefore, the latter is usually used to analyse the Eu XPS spectra and better estimate the Eu initial valence [34,35,36,37].

Figure 6a–d shows the Eu 3*d* XPS spectra after subtracting an optimised Shirley background, measured at *t* =210 s, 360 s, 450 s and 510 s. The peaks were deconvoluted using Gaussian–Lorentzian fitting, while the $\chi^{2}$ value indicates the quality of the fit. Although the Eu${}_{3}$O${}_{4}$ layer was etched fully, only these scans were considered for the analysis of the Eu cation valency (the yellow shaded area of Figure 5a) to minimise the effect of interdiffusion at the SiO${}_{2}$/Eu${}_{3}$O${}_{4}$ and Eu${}_{3}$O${}_{4}$/Au interfaces and increase the intensity of the Eu 3*d* and 4*d* peaks.

All spectra in Figure 6 show the spin-orbit coupling (SOC) components, 3*d*${}_{5/2}$ and 3*d*${}_{3/2}$, for Eu${}^{2+}$ and Eu${}^{3+}$ separated by Δ∼29.5 eV, which agrees with previously reported values [34,36,38]. They also show additional peaks at slightly higher binding energy (BE) to the SOC peaks for the Eu${}^{2+}$ and Eu${}^{3+}$. These shake-up satellite peaks arise as a result of the multiplet structures of the 4*f*${}^{7}-3$*d* hole in the final state [37]. Furthermore, the fast 3*d* photoelectrons create plasmon excitation structures observed as broad peaks at BE ∼1146 eV and ∼1170 eV [34]. The XPS spectra shown in Figure 6 prove the mixed valency of Eu cations as they agree well with previous work reported for Eu${}^{2+}$ and Eu${}^{3+}$ [34,36,39,40,41]. Moreover, the average atomic ratio of the Eu${}^{2+}$ to Eu${}^{3+}$ in the 3*d*${}_{5/2}$ and 3*d*${}_{3/2}$∼28:72 is consistent with the values reported in Eu-doped ZnO [42] and Eu-doped GaN nanowires [43]. Table 1 summarises the positions of the Eu${}^{2+}$ and Eu${}^{3+}$ 3*d* peaks, their FWHM, their corresponding multiplet satellites, the ratio of Eu${}^{2+}$/Eu${}^{3+}$ and the fits $\chi^{2}$ values of the four spectra.

### 3.4. Raman Spectroscopy

Raman spectroscopy is a versatile and nondestructive technique widely used to study the graphene’s structural and electronic properties [44,45]. A good-quality monolayer of graphene has two main characteristic Raman peaks; the *G* and 2*D* peaks at ∼1582 cm${}^{-1}$ and ∼2700 cm${}^{-1}$, respectively. It can also possess other disorder-induced peaks, such as the *D* peak at ∼1350 cm${}^{-1}$ [46,47,48]. Therefore, the presence or absence of these peaks and the ratio of the intensity of the *D* peak to the intensity of the *G* peak (*I*${}_{D}$/*I*${}_{G}$), which reflects the defect density in the graphene structure, were used to assess the quality of our graphene underlayer [49].

Figure 7a,b show the microscopic optical images of the sample highlighting three different regions of the Si/SiO${}_{2}$/graphene/Eu${}_{3}$O${}_{4}$/Au sample taken before etching the Au capping layer. The zoomed-in image collected with a ×100 objective lens (Figure 7b) shows that the graphene layer consists of discontinuous mixture of mono- and multilayer graphene domains rather than a continuous homogeneous monolayer, which could be either a result of the growth of the Eu${}_{3}$O${}_{4}$ film or the quality of the commercial graphene. Therefore, one would expect the presence of defect-induced peaks in the Raman scans [49].

Figure 7c,d show the microscopic optical images of the graphene edge under the Eu${}_{3}$O${}_{4}$ film before and after removing the Au layer, respectively. No significant change in the contrast is observed between the two images, suggesting that the etching process did not remove the graphene underlayer.

Four random Raman scans taking on two different areas of the sample’s surface, bare graphene and the graphene/Eu${}_{3}$O${}_{4}$ region after removing the Au layer, are shown in Figure 8a,b, respectively. The emergence of the additional defect-induced peaks in the spectra of the graphene/Eu${}_{3}$O${}_{4}$ area and the increase in their intensities (Figure 8b) compared to the scans of the bare graphene (Figure 8a) indicates that the growth of Eu${}_{3}$O${}_{4}$ film increased the defect density in the graphene structure. This is also seen by the shift in the *I*${}_{D}$/*I*${}_{G}$ ratio of the graphene under the Eu${}_{3}$O${}_{4}$ film towards the higher values (Figure 9b), compared to the bare graphene sheet (Figure 9a). This is because *I*${}_{D}$/*I*${}_{G}$ is known to be small for low-defect-density graphene [48,50]. However, the optical images (Figure 7) and Raman measurements (Figure 8) show that the underlayer graphene is largely maintained after the growth of the Eu${}_{3}$O${}_{4}$ film. Moreover, the data suggest the growth parameters can be optimised further to reduce the effect of the deposition of Eu${}_{3}$O${}_{4}$ and improve the Eu${}_{3}$O${}_{4}$/graphene interface.

## 4. Conclusions

In summary, we discussed the experimental work carried out to study the growth of Eu${}_{3}$O${}_{4}$ thin film on Si/SiO${}_{2}$ and on a graphene sheet by MBE at 400${}^{\circ }$C. The structural and magnetic characterisations show successful deposition of crystalline, highly textured Eu${}_{3}$O${}_{4}$(001) films with a *T*_C_ of ∼5.5±0.1 K and a magnetic moment of ∼320 emu/cm${}^{3}$ at 2 K. However, the films show no metamagnetic behaviour which could be attributed to the strain from the substrate. Furthermore, a qualitative analysis of the depth-profile XPS scans confirms the mixed valency of the Eu cation with the ${\mathrm{Eu}}^{2+}:{\mathrm{Eu}}^{3+}$ ratio of 28:72, which agrees with the values reported for other Eu-doped systems. Optical microscopy and Raman measurements show that the graphene layer remains largely intact after the Eu${}_{3}$O${}_{4}$ growth. Therefore, this study represents the first successful step towards integrating a Eu${}_{3}$O${}_{4}$ thin film in two widely used electronic substrates for future spintronics applications.

## Figures and Tables

**Figure 1 nanomaterials-11-01598-f001:**
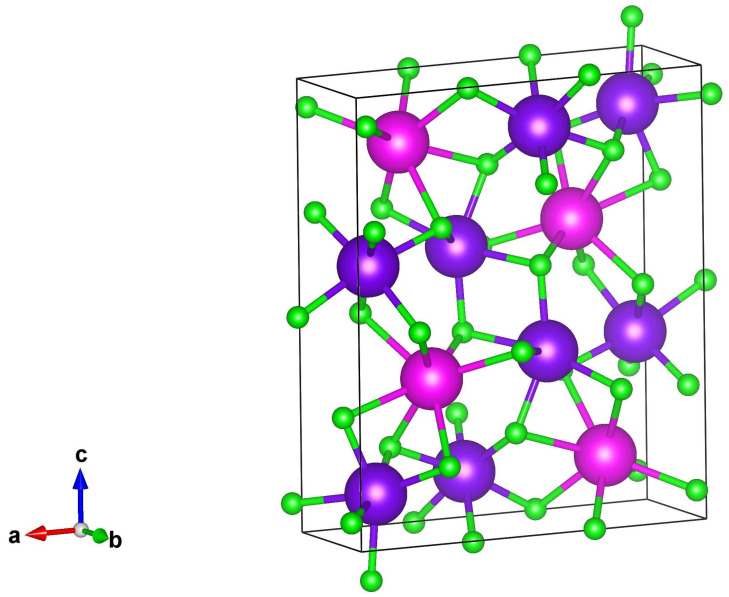
The orthorhombic structure of Eu${}_{3}$O${}_{4}$, in which the purple, magenta and green spheres represent the Eu${}^{3+}$, Eu${}^{2+}$ and O${}^{2-}$ ions, respectively. The black box represents one unit cell of the Eu${}_{3}$O${}_{4}$ crystal. The model was constructed using VESTA^*TM*^ software package [6].

**Figure 2 nanomaterials-11-01598-f002:**
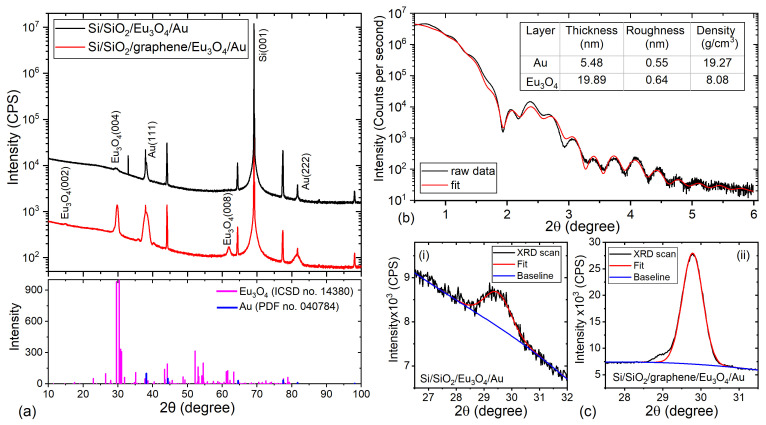
(**a**) The RT XRD scan of the Si/SiO${}_{2}$/Eu${}_{3}$O${}_{4}$(001)/Au (black line) and Si/SiO${}_{2}$/graphene/Eu${}_{3}$O${}_{4}$(001)/ Au (red line) samples carried out between 10${}^{\circ }$ and 100${}^{\circ }$ using a monochromator and a 1D detector. The scan for the Si/SiO${}_{2}$/graphene/Eu${}_{3}$O${}_{4}$(001)/Au sample is downshifted by a factor of five for ease of comparison. The bottom panel shows the reference patterns for Eu${}_{3}$O${}_{4}$ (magenta—ICSD no. 14380) and Au (blue—PDF no. 040784). (**b**) The XRR measurement of the Si/SiO${}_{2}$/Eu${}_{3}$O${}_{4}$(001)/Au sample (black line) and the corresponding fit (red line) between 0${}^{\circ }$ and 6${}^{\circ }$. The table lists the thickness, roughness and density of the deposited films as deduced from the fit. (**c**) The Gaussian fit for the Eu${}_{3}$O${}_{4}$(004) peak of the Si/SiO${}_{2}$/Eu${}_{3}$O${}_{4}$(001)/Au (i) and the Si/SiO${}_{2}$/graphene/Eu${}_{3}$O${}_{4}$(001)/Au sample (ii).

**Figure 3 nanomaterials-11-01598-f003:**
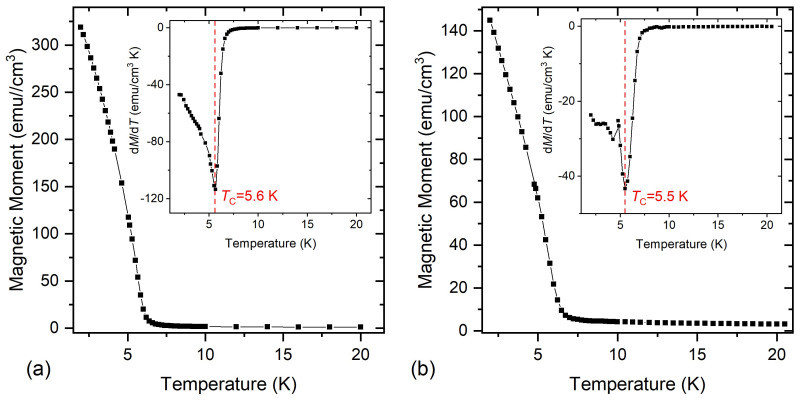
(**a**) The 20 Oe FC *M* vs. *T* measurement for Si/SiO${}_{2}$/Eu${}_{3}$O${}_{4}$/Au. The inset presents the d*M*/d*T* vs. *T* plot used to determine the *T*_C_. (**b**) The ZFC *M* vs. *T* measurement for the Si/SiO${}_{2}$/graphene/Eu${}_{3}$O${}_{4}$/Au sample. The inset shows the d*M*/d*T* vs. *T* graph used to deduce the *T*_C_ of the Eu${}_{3}$O${}_{4}$.

**Figure 4 nanomaterials-11-01598-f004:**
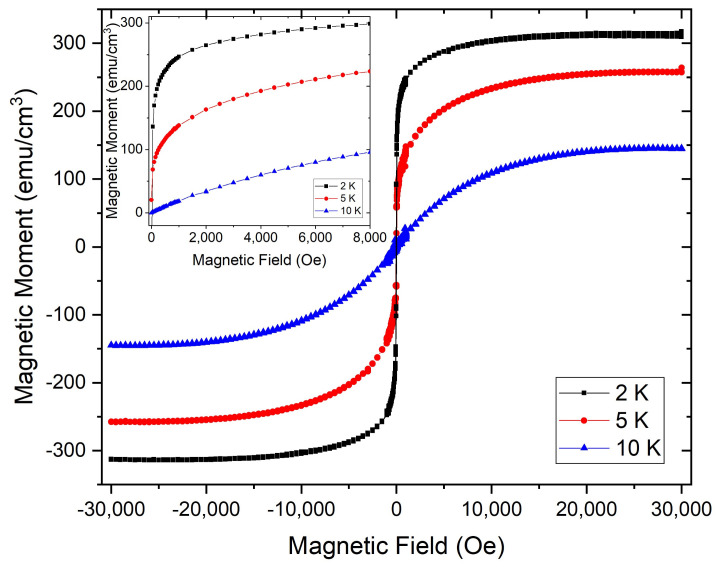
ZFC Isothermal magnetisation hysteresis loops of the Si/SiO${}_{2}$/graphene/Eu${}_{3}$O${}_{4}$/Au sample measured as a function of the applied in-plane magnetic field at 2 K, 5 K and 10 K. The inset highlights the virgin magnetisation curves at these temperatures.

**Figure 5 nanomaterials-11-01598-f005:**
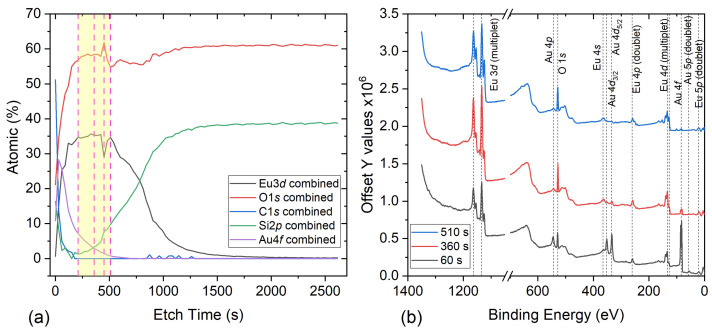
(**a**) XPS etch profile of the Si/SiO${}_{2}$/Eu${}_{3}$O${}_{4}$/Au sample by Ar${}^{+}$ plasma etching. The yellow shaded region highlights the area considered for the analysis of Eu cations, whereas the dashed vertical lines indicate t= 210 s, 360 s, 450 s and 510 s. (**b**) The XPS survey spectra of the Si/SiO${}_{2}$/Eu${}_{3}$O${}_{4}$/Au sample, collected at *t*
=60 s, *t*
=360 s and *t*
=510 s, highlighting the Au, Eu and O peaks.

**Figure 6 nanomaterials-11-01598-f006:**
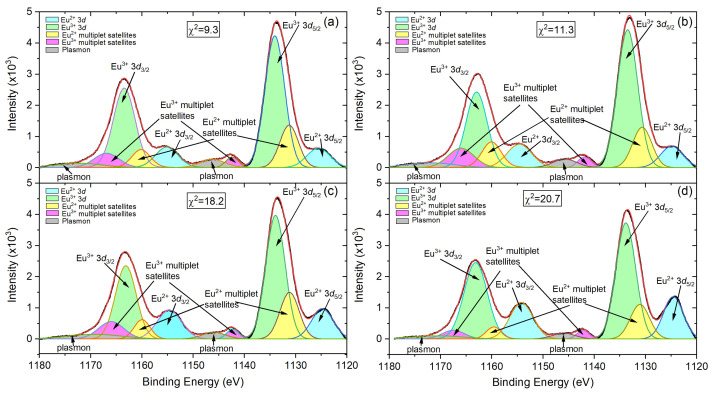
The deconvoluted 3*d* XPS spectra of the Eu${}_{3}$O${}_{4}$ film on Si/SiO${}_{2}$ substrate using Al Kα source (1486.68 eV) after the Shirley background subtraction, measured at etching time (**a**) *t* = 210 s, (**b**) *t* = 360 s, (**c**) *t* = 450 s and (**d**) *t* = 510 s. The raw data (black line), fitting curve (red line), Eu${}^{2+}$ 3*d* (blue shaded peaks), Eu${}^{2+}$ multiplet satellites (yellow shaded peaks), Eu${}^{3+}$3*d* (green shaded peaks), Eu${}^{3+}$ multiplet satellites (magenta shaded peaks) and plasmon excitations (grey shaded peaks).

**Figure 7 nanomaterials-11-01598-f007:**
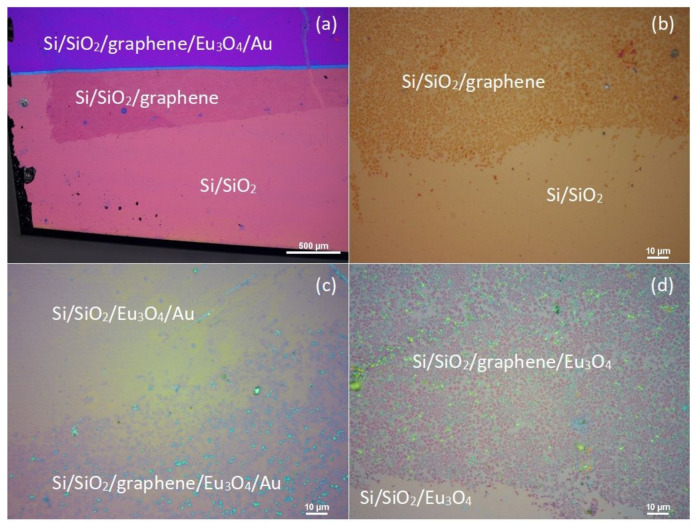
The microscopic optical images of the Si/SiO${}_{2}$/graphene/Eu${}_{3}$O${}_{4}$/Au sample highlighting the different regions of the surface of the sample. (**a**) the bare and coated area of the Si/SiO${}_{2}$ substrate before the etching process, (**b**) a zoomed-in view of the surface using a 100X objective lens showing the mixed domain structures of the graphene underlayer, (**c**) the graphene under the Eu${}_{3}$O${}_{4}$ layer before and (**d**) after etching the Au capping layer.

**Figure 8 nanomaterials-11-01598-f008:**
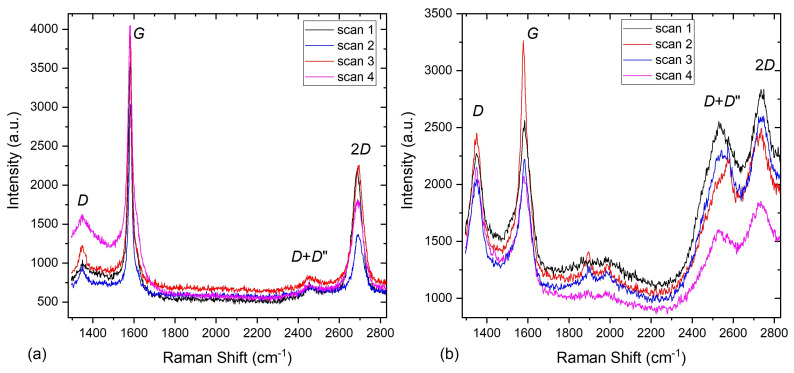
RT Raman spectroscopy measurements for (**a**) the bare graphene sheet on the Si/SiO${}_{2}$ substrate and (**b**) graphene under the Eu${}_{3}$O${}_{4}$ film after etching the Au capping layer.

**Figure 9 nanomaterials-11-01598-f009:**
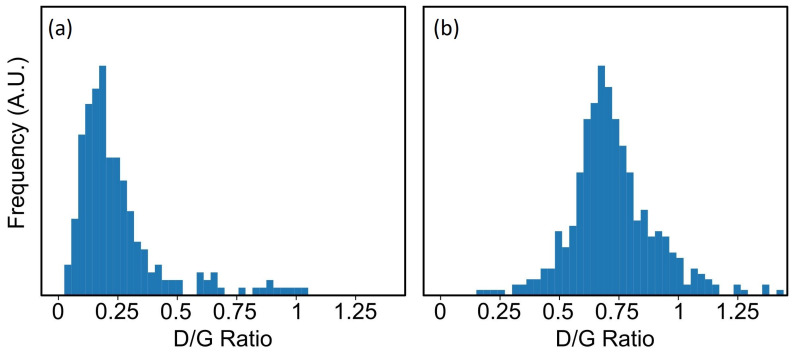
The frequency distribution of the ratio of the intensity of the *D* peak to that of the *G* peak of (**a**) the bare graphene and (**b**) the graphene under the Eu${}_{3}$O${}_{4}$ film. Raman spectra of the regions with no graphene signal were removed from the statistics.

**Table 1 nanomaterials-11-01598-t001:** The positions of the Eu${}^{2+}$ and Eu${}^{3+}$ 3*d* peaks, their FWHM and the position of their corresponding multiplet satellites. The atomic ratios of the Eu${}^{2+}$ to Eu${}^{3+}$ were deduced from the areas of the peaks. The table also lists the $\chi^{2}$ value of the fittings.

Spectrum	3*d*${}_{5/2}$	FWHM	3*d*${}_{5/2}$ Satellites	3*d*${}_{3/2}$	FWHM	3*d*${}_{3/2}$ Satellites
**Eu** ${}^{2+}$ **(eV)**						
*t*=210 s	1125.54	5.39	1131.26	1155.14	5.29	1160.16
*t*=360 s	1124.88	5.17	1130.64	1154.75	5.76	1160.04
*t*=450 s	1124.78	5.01	1131.19	1154.65	5.64	1159.96
*t*=410 s	1124.56	4.58	1131.07	1154.21	5.36	1159.58
**Eu** ${}^{3+}$ **(eV)**						
*t*=210 s	1134.07	4.56	1142.39	1163.46	5.03	1166.92
*t*=360 s	1133.49	4.69	1141.90	1162.96	4.85	1166.00
*t*=450 s	1133.93	4.46	1142.29	1163.15	4.80	1166.00
*t*=410 s	1133.83	4.56	1142.23	1163.15	5.38	1167.48
	**Atomic ratio 3** ***d*** ${}_{5/2}$			**Atomic ratio 3** ***d*** ${}_{3/2}$		$\chi^{2}$
	**Eu** ${}^{2+}$	**Eu** ${}^{3+}$		**Eu** ${}^{2+}$	**Eu** ${}^{3+}$	
*t*=210 s	49.81	50.19		21.39	78.61	9.30
*t*=360 s	14.63	85.37		26.84	73.16	11.30
*t*=450 s	21.58	78.42		31.61	68.39	18.20
*t*=410 s	26.80	73.20		31.96	68.04	20.70
Average	28.20	71.80		27.95	72.05	

## Data Availability

The data presented in this study are available from the corresponding author upon request.

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
