# Peer review of "Growth and Characterisation Studies of Eu3O4 Thin Films Grown on Si/SiO2 and Graphene"

_nanomaterials, 2021, doi:10.3390/nano11061598_

Round 1
Reviewer 1 Report
The present manuscript entitled “GROWTH AND CHARACTERISATION STUDIES OF Eu3O4 THIN FILMS GROWN ON Si/SiO2 AND GRAPHENE” authored by Aboljadayel et al. describes the growth, structural and magnetic properties of the Eu-oxide phase, Eu3O4, thin films grown on a Si/SiO2 substrate and Si/SiO2/graphene using molecular beam epitaxy. It is a well-organized article and lacks major errors, and thus, I recommend it for publication. However, certain Minor issues are detailed below which need to be addressed before its final acceptance in the Nanomaterials.
I advise the authors to take the following points into account while revising their manuscript.
Comment 1: Firstly, I would like to draw attention of the authors that I found there are so many typographical errors in the manuscript, so authors need to correct it in the revised manuscript.
Comment 2: Moderate English changes required in the manuscript.
Comment 3: Some relevant references in this area are still missing in introduction section, so include some significant relevant references from the recent years.
Comment 4: The similarity report of the article is *13%*. However, the authors need to rewrite some part of introduction section (Line 38 to 53). So, find the attached similarity report for corrections.
Comment 5: Line 91 “The RT XRD scans (from 20° - 100°)” should be “The RT XRD scans (from 10° - 100°)”.
Comment 6: Improve the quality Figure 1, which is currently quite disorganized.
Comment 7: In XRD results: The authors should explore and discuss better their results with some more references to prepare a better discussion.
Comment 8: In Figure 7, the scale bar is not properly visible. So, redraw the scale bar manually.
Comment 9: In Figure 9 image, there is no mentioning of (a) and (b). So, add the details of (a) and (b) the Figure 9.
Comment 10: The conclusions section is too short, the authors should revise it.

Reviewer 2 Report
This is an interesting manuscript on the growth of Eu3O4 thin films on two types of substrates: Si/SIO2 and Si/SiO2/ graphene, resepctively. These films exhibit ferrmagnetism at low temperature, with nometamagnetic order, due to existence of strain, as concluded by this study. As a suggestion, the XRD studies could be followed by a strain analysis by using the Williamson-HAll formalism to assess, both qualitatively and quantitatively, the existence of strain in these films. The oxidation states of Eu ions was studied by a comprehensive XPs study, the authors finding that Eu ions are in a mixed valence state. The manuscript is well written with some minor typos (see, for example LIne 108 on Page 4) and the discussion and conclusions are well supported by the experimental data. Based upon foregoing, I recommend that the manuscript be accepted for publication in Nanomaterials.
Reviewer 3 Report
In my opinion presented work is good, at the whole volume.
Author Response
Thank you. We appreciate the time and effort you dedicated to providing feedback on our manuscript. We have revised the whole document and corrected the typos, which are marked up using the "track changes" function.
Reviewer 4 Report
The manuscript reports on growth and methods of characterization of Eu3O4 thin films grown on silicon/silicon oxide and graphene substrates. The subject is within the scope of MDPI Nanomaterials. The authors have carried out X-Ray diffraction scans, SQUID and XPS measurements as well as Raman spectroscopy studies of the examined samples. The paper is purely focused on experimental work.
The paper is well written and it deserves publication since it includes a comprehensive study of compounds based on a less known chemical element i.e. europium. However it is felt that the authors might enrich the manuscript by writing in more detail on the potential applications of the examined thin films for spintronics. In particular it would be interesting to learn of the advantages to use europium instead of other lanthanoids, since some of them are more abundant in Nature or available on market than their more “exotic” counterpart.
Author Response
Comment 1: the authors might enrich the manuscript by writing in more detail on the potential applications of the examined thin films for spintronics. In particular it would be interesting to learn of the advantages to use europium instead of other lanthanoids, since some of them are more abundant in Nature or available on market than their more “exotic” counterpart.
Response: Thank you for giving us the opportunity to submit a revised draft of the manuscript. We also appreciate the time and effort you dedicated to providing feedback on our manuscript and are grateful for the insightful comments and valuable suggestions for improving our paper.
We extended the introduction by briefly discussing the applications of Eu chalcogenides in general and focusing on the examined Eu3O4 compound and the well-studied Eu-oxide phase, EuO. Changes are marked up using the “track changes” function.